# Perception Towards Organic vs. Conventional Products in Romania

**Vasile Stoleru *****, Neculai Munteanu and Andrei Istrate**

Faculty of Horticulture, University of Agriculture Sciences and Veterinary Medicine, 3 M. Sadoveanu, 700440 Iasi, Romania; nmunte@uaiasi.ro (N.M.); andrei.istrate87@gmail.com (A.I.)

* Correspondence: vstoleru@uaiasi.ro; Tel.: +40-232-407530

**Abstract:** The aim of this study was to elicit answers referring to the consumer perception with respect to organic products. Factors that determine behavior were also considered: Gender, age, education, income, or social status. Analysis of data collected revealed that perception is the psycho-cognitive element that may determine the expression of behavior in relation to the organic production system. Furthermore, organic farming in Romania is a relatively recently formed market segment. The study was carried out by using a questionnaire developed specifically for this purpose, on a sample of 226 respondents. The data obtained from the survey were analyzed by employing the contingency coefficient and Pearson chi-square tests, using the SPSS software version 20. The perception of organic food is associated with its nutritional quality or sensory attributes (appearance, taste, flavor).

**Keywords:** organic and non-organic food; perception; consumer

## 1. Introduction

Agricultural products must always be present in human nutrition, as they are indispensable for equilibrated diets since they discharge dietary fiber, phytochemicals, vitamins, and minerals [1]. Although Romania has significant potential to increase organic farming, this is determined by subsidies from its common agricultural policy and free export of organic products to the countries of the European Union [2,3].

After an oscillating evolution of organic land 2000–2014, the current area is situated at 259,000 ha, representing some 0.31% of the total agricultural area [4]. The total yield in the same period was at the level of 1.1 million tons.

From this total, Romania exported over 175,000 tons of organic foods with a value of more than 200 million € in 2015. Sales grew equally by another 20% in 2012 compared to 2017 [5]. The increase in exports is also due to the fact that from July 2010 onwards, all preplaced organic products produced within the EU must carry the new logo [6,7]. A key factor for organic agriculture is the perception of consumers related to the organic products, in terms of attitude and preferences, as particular expressions of their behavior.

Consumer behavior refers specifically to the actions taken by the consumer when deciding if they should buy or not buy a product. The actual buying decision is not taken instantly in most cases but appears to be the result of a sequential process with a certain duration of time, determined by economic, social, and cultural factors [8]. In cognitive terms, perception can be defined as the optional ability to understand the phenomena of the external world, namely knowledge [9,10].

When analyzing consumer behavior, one should consider the following traits: What consumers think (perception), feel (feeling), and do (behavior), alongside factors that influence them (environment) [11,12]. Consumer behavior can hence be influenced by experience; this can lead to a change in attitudes and behavior [13,14]. The factors that determine consumer perception refer not

only to physical needs (food, health, environmental protection, etc.) [15] but also their dependency on other socio-cultural components: Culture, religion, training, income, and social position [16,17].

The family is probably the first form of a coherently organized and institutionalized society which provides the conditions to meet the basic needs of individuals and human societies: Sexuality, procreation, economic survival, personal and collective identification, raising descendants, and education [8,18]. Generally, affiliation to a certain social class is determined by an individual's occupation. People with similar incomes but different education will spend money in very different ways [10,19].

Before a purchase happens, people must be aware of the product. Awareness is a process that begins with exposure to sensory stimuli representing the product and continues by paying attention, understanding, acceptance, and retention of information to form a mental representation of the object [18].

This is not always necessary, particularly when new products are offered at a marketplace and consumers have no experience with it or knowledge of it. Before a purchase, many attributes of a food product cannot be revealed, so consumers develop expectations about quality when selecting it [20]. In this case, expectations of new products are framed by the information on the label and previous experience with a similar product [21].

This also involves associating the stimulus to some verbal concepts—expensive, cheap, durable, and economical—or images [9]. The sale price is also associated in some cases with a certain quality of a product, providing confidence in the purchase decision.

In this respect, some general knowledge about what an organic producer means and how this product is obtained is very important in decision making.

Organic farming maintains and increases soil fertility through natural resources and fertilizers rather than using chemical fertilizers. For the soil to retain fertility, farmers use green manure, such as oats, vetches, and legumes. Growing plants for green manure is expensive, and biological production is ploughed back into the biological ecosystem circuit and not sold [2]. For producers, it is important to know the preferences of consumers to adapt their products for the needs of the consumer [22]. These features are: The quality of the food, the presentation of the assortment, design, location, and method of distribution, dependability of the farmer or manufacturer, brand image and product category, and the label for organic products and certification bodies as a guarantee of food safety for the products [23].

Placement is often part of the product, meaning that there is a specific benefit to the distribution of the utility space. Marketers need to consider the balance between the usefulness of commercial space (cognitive perception) and pleasant perception of hedonistic characteristics as affective aspects [24].

Sensory and organoleptic attributes, experienced directly by consumers, include size, color, form, taste, smell, and 'feel'. These may, however, be of relatively peripheral significance since there is no guarantee that the food had been produced organically just because it smelled good or tasted differently [25,26].

Consumers were unable to assess the quality of organic food simply from its physical characteristics but needed reassurance from credible industry standards that aided the perception of 'extrinsic' quality and also the safety of organic foods. So, their confidence for the product quality is extremely important and is based on specific certification [27]. Certification, if working properly, should incorporate the perception of organic quality as a symbol of sustainable agriculture and healthy living, interwoven with process-related quality as well as the use of safe or natural raw materials. Indications existed that such attitudes were encouraged by a lack of faith in the conventional food sector [22,28].

Concerns related to the environment are evident in the increasingly environmentally conscious marketplace [12]. Over the years, a majority of consumers have realized that their purchasing behavior has a direct impact on many ecological problems [29].

Concerning the presented motivation, the aim of this study was to elicit some needed answers to be used in both the measurement, but also the segmentation, of the perception of the consumers with respect to going organic.

## 2. Materials and Methods

### 2.1. Population and Sample

The study was applied on a sample of 226 persons, structured according to demographics and acknowledgment that the trend in Romanian consumption of organic produce is determined mainly by the urban population.

The determining sample is an essential activity which favors the survey quality, and this study strived to achieve and reflect the "local mix".

The respondents were selected from a range of 20 to 49 years as follows: 33.6% belonging to the 20–29 age group, 35% to the 30–39 age group, and 31.4% to the 40–49 age group.

The sample was chosen from frequented places, since people of all ages come to relax in these locations, do not rush, and the questionnaire can be completed without the respondent being distracted in any way. The questionnaire was assisted for uniformity purposes.

### 2.2. Instrumentation

The research method employed was a quantitative study using a self-questionnaire, delivered face to face by the interviewers, between March and May 2013. We note that an average of 26 surveys/day were completed, between 12:00–16:00 h or 16:00–18:00 h. in the main urban centers of Romania (Iasi, Bucharest, Timisoara, Cluj, and Constanta).

The questionnaire included a list of questions in order to collect information from the target groups within the sample interviewed [11,30,31].

In order to develop consumer perception for organic products, we developed two series of questionnaires aimed at identifying the perceived differences between organic and conventional products as well as the buying behavior for these products.

The present questionnaire used closed questions with presented answers, as this made it much easier for the responders to choose.

Drafting the questionnaire was one of the most challenging parts of the study since the following needed to be taken into consideration: The administration of the questionnaire, which influenced the length of the questionnaire; the types of question, which determined how data would be processed; and the language used, since the respondents were from different social classes, etc.

The questionnaire consisted of 14 problems, in three sections: (1) Filtering questions, (2) differences in perception between organic and conventional foods (common), and (3) socio-professional characteristics. The first part of the questionnaire focused on sample filters: Gender, age of the respondent (20–29 years, 30–39 years, 40–49 years), and regularity of consumption.

The second group of questions was aimed at identifying the perceived differences between organic and conventional produce such as:

- Compared conventional products with organic products are: Less tasty, just as tasty, tastier, less flavorful, just as flavorful, more flavorful, fast breaks, spoil as quickly, less healthy, just as healthy, healthier, less appealing, appealing, more appealing. For questions about consumer perception regarding health, taste, and appearance, we used a Likert scale with three response options (Less ... = 1; Just of ... = 2; More ... = 3), which created a homogeneous symmetry between the three types of response.
- Differences perceived by consumers between organic and conventional products for: Content of chemicals, healthy, environmental protection, price, preservation, label, appearance, absence of genetically modified organisms (GMOs) etc. At this question, consumers had to choose the most important criterion for differentiating between organic and conventional products.

The third part of the questionnaire held demographic and profession questions, which, in the authors' opinion, determine consumer perception: Household size; marital status; the educational level of the respondents (without education, primary/secondary school, vocational school, high school,

college/university study); monthly income per family member (less than 400 €, 401–800 €, 801–1200 €, 1201–1600 €, more than 1600 €); house ownership: Full ownership or rental.

## 2.3. Validation of the Questionnaire

Whatever the experience in the design of the questionnaire, it is necessary to carry out pre-survey tests by checking for the understanding, interpretation, and acceptability of the questions [11,30].

Testing the questionnaire is essential in any situation and any type of survey, knowing that consumers generally have different levels of training or do not have a specialized technical language. In studies about opinions, motivations, or intentions, questionnaire development is more difficult, and questions will be selected and studied carefully. In these situations, questions that lead to the occurrence of errors in responses should be avoided as much as possible. Some errors can be caused voluntarily by the respondents while attempting to mask attitudes and opinions or may be unintentional due to a misunderstanding of the issues and semantics.

The questionnaire was tested on 20 persons from different age groups with different educational levels. During this phase, the following aspects were considered: Ease of understanding, ease of collecting the responses, the questions and their length, the degree of coverage of the categories chosen, bias avoidance, etc.

## 2.4. Data Analysis

Data were quantitatively collected and analyzed using the SPSS software version 20 from IBM. A variety of descriptive and inferential statistics were utilized to understand significant differences in consumer perception according to the demographics for specific attributes.

The ANOVA test was utilized to summarize the demographic profile. For the descriptive statistics, contingency coefficient (C) and Person chi-square ($\chi^2$) tests were used for consumer perception, for $p < 0.05$, thus demonstrating the dependence of the variables, or the probability that any observed difference between variables will occur by chance.

To test for differences in means of comparative consumer perception between organic and conventional systems, several constructs, based on consistency in content, were established from the items on the questionnaire.

## 2.5. The Demographic Profile

The demographic profile of the interviewers presented in Table 1 shows that 118 were female (52.2%) and 108 were male (47.8%).

Regarding the surveyed age, it can be said that the research tool was applied to active persons in an urban area, at least 20 years old, regardless of education, culture, ethnicity, and age group. Thus, the 20–29-year-olds amounted to 76 people (33.6%), 35.0% were from the age group 30–39 years, and the 40–49-year-olds group included 71 respondents, which is 31.4%.

The sample of questionnaires was chosen taking into account the fact that people aged 20–29 years old across the entire population consist of 48% men and 52% women, which makes the selected group close to these values.

The social status of people is a demographic criterion that may or may not depend on the attitude of the respondents at the family level. Of the total respondents, 52.7% were married, 39.8% were unmarried, 5.8% were divorced, and 1.8% were widowed. The higher number of unmarried people is explained by the fact that the sample was carried out in cities which are university centers with a high urban population.

Concerning the educational level, it can be said that of the 226 surveyed, 3 were without any education (1.3%), 3 people had primary/secondary school (1.3%), 26 persons had vocational school (11.5%), 70 of the respondents attended high school (31%), and 124 persons, 54.9% of the sample, were graduates or attended university courses, which did not deviate greatly from the population structure.

The respondent's income level was calculated as the monthly average for each family member over the last months. Of the total respondents, 58 earned less than 400 €/month (25.7%), 71 people between 401–800 € (31.4%), 41 between 801–1200 € (18.1%), 26 persons drew 1201–1600 € (11.5%), and 30 respondents earned more than 1600 € (13.3%). Comparing the income of the sample and the population structure, it can be said that the selected sample was representative of the study.

**Table 1.** Demographic characteristics of respondents.

| Variable and Level | Number of Interviews | Relative Frequency (%) | Cumulative Percentage (%) | Standard Deviation | Variance |
|---|---|---|---|---|---|
| Gender | 226 | 100 | 100 | | |
| male | 108 | 47.8 | 47.8 | 0.501 | 0.251 |
| female | 118 | 52.2 | 100 | | |
| Age | 226 | 100 | 100 | | |
| 20–29 years | 76 | 33.6 | 33.6 | | |
| 30–39 years | 79 | 35.0 | 68.6 | 0.808 | 0.653 |
| 40–49 years | 71 | 31.4 | 100 | | |
| Marital status | 226 | 100 | 100 | | |
| unmarried | 90 | 39.8 | 39.8 | | |
| married | 119 | 52.7 | 92.5 | 0.660 | 0.435 |
| divorced | 13 | 5.8 | 98.2 | | |
| widowed | 4 | 1.8 | 100 | | |
| Educational level | 226 | 100 | 100 | | |
| no education | 3 | 1.3 | 1.3 | | |
| primary/secondary school | 3 | 1.3 | 2.7 | 0.839 | 0.705 |
| vocational school | 26 | 11.5 | 14.2 | | |
| high school | 70 | 31.0 | 45.1 | | |
| university study | 124 | 54.9 | 100.0 | | |
| Income * | 226 | 100 | 100 | | |
| under 400 € | 58 | 25.7 | 25.7 | | |
| 401–800 € | 71 | 31.4 | 57.1 | | |
| 801–1200 € | 41 | 18.1 | 75.2 | 1.340 | 1.795 |
| 1201–1600 € | 26 | 11.5 | 86.7 | | |
| more than 1600 € | 30 | 13.3 | 100.0 | | |

* The income is calculated as the average per family member.

## 3. Results

### 3.1. The Criteria for Identification of the Differences Perceived by Consumers

The study of consumer perception towards organic products is associated primarily with the default quality of these products.

The perceptual aspect of the respondents was tested, dependent on the demographic profile of the respondents according to gender, age, education level, and income (Table 2). The most mentioned differences that organic products have, according to the gender of the respondents, were related to healthy benefits (25.66%), less on synthetic chemicals (19.91%), tastier (16.37%), as well as technology and environmental protection used for organic products (10.62%). In addition, quite a large number of people (11.50%) did not perceive differences between organic and conventional products. The other criteria of differentiation between organic and conventional products are linked to the price (6.63%), appearance (3.53%), label (2.21%), the validity of organic products (1.76%), and non GMO (1.33%), regardless of the respondent's gender.

According to age, 31.57% of the 20–29 years old age group believe that organic is healthier, has a lower content of chemicals (21.05%), and is tastier (15.79%). In addition, 15.79% of this age group do not know the difference between organic and conventional products, which means they do not have sufficient knowledge to detect differences between the categories. A different segmentation of the

responses is observed for the age group 30–39 years, in which 22.78% of the responses converge to the perception that organic products are free of synthetic chemicals, 18.98% believe that products are tastier and healthier, and 16.46% of the cases consider that production technology and environmental protection is what makes the difference between those two categories. Not to be overlooked is the fact that this age group is better informed about the variances between the certified and conventional organic products, at 94.94%. The 40–49-years age group gave the expected answers, meaning that 26.76% of the respondents consider organic the healthiest; otherwise, the answers are segmented almost homogeneously on the item level, between 8–11 answers. The level of education was a main element of perception between products and their nutritional quality. Regarding the people with a technical background, a relatively homogeneous segmentation of the categories from the items mentioned was observed. The respondents with a high school education mentioned in 27.14% of the cases the elements of classification between the two categories, the nutritional quality, and for 25.71% of the persons, taste was an important element of creating perception. The smaller content of synthetic chemical, production technology, and price are other important criteria of differentiation. The people with university education provided answers as expected, in the sense that for 50% of them, being qualitatively low in chemical components is the main criterion for the differentiation, in which organic products can be consumed rather than conventional products. In addition, organic technology, taste, the price, and label are criteria to be taken into account in determining consumers' perceptions according to the certified products [19].

Not to be overlooked is the fact that 10.48% of the people with university education do not perceive the differences between organic and chemically treated products. In terms of perception, regarding the level of income, 58% of the respondents consider the nutritional quality of untreated products a really important factor, especially people with an income less than 1200 € or more than 1600 €. The people with an income between 1200 € and 1800 € considered the segmentation between organic and conventional products being low in chemical content to be the main criterion. In addition, taste (16.37%) and the technology for organic products (10.62%) are important criteria by which the interviewees perceive as the differences between the products. The data presented in Table 2, although numerically and percentage different, are not significant because $p > 0.05$, which shows that there is no dependence between perception and socio-professional profile.

**Table 2.** Differences perceived by respondents between organic and conventional products.

| Variable and Level | Do not Know | Tasty | Synthetic Chemical Content | More Healthy | Environment Protection | Price | Appea -Rance | Preserva Tion | Label | Non GMO | Total | C (Sign) | $\chi^2$ (Sign) |
|---|---|---|---|---|---|---|---|---|---|---|---|---|---|
| Gender | | | | | | | | | | | 226 | | |
| male | 16 | 15 | 22 | 29 | 12 | 8 | 7 | 2 | 5 | 2 | 118 | 0.246 | 14.551 |
| female | 10 | 22 | 23 | 29 | 12 | 7 | 1 | 2 | 0 | 1 | 108 | (0.267) [a] | (0.267) [a] |
| Age | | | | | | | | | | | 226 | | |
| 20–29 years | 12 | 12 | 16 | 24 | 3 | 3 | 2 | 1 | 1 | 2 | 76 | 0.347 | 30.921 |
| 30–39 years | 4 | 15 | 18 | 15 | 13 | 4 | 5 | 2 | 2 | 1 | 79 | (0.156) [a] | (0.156) [a] |
| 40–49 years | 10 | 10 | 11 | 19 | 8 | 8 | 1 | 1 | 2 | 1 | 71 | | |
| Educ. level | | | | | | | | | | | 226 | | |
| no education | 0 | 0 | 0 | 2 | 0 | 1 | 0 | 0 | 0 | 0 | 3 | | |
| primary/2nd school | 1 | 0 | 0 | 2 | 0 | 0 | 0 | 0 | 0 | 0 | 3 | 0.413 | 46.394 |
| vocational school | 4 | 7 | 5 | 4 | 1 | 2 | 2 | 1 | 0 | 0 | 26 | (0.539) [a] | (0.539) [a] |
| high school | 8 | 18 | 9 | 19 | 6 | 6 | 2 | 0 | 1 | 1 | 70 | | |
| university studies | 13 | 12 | 31 | 31 | 17 | 6 | 4 | 3 | 5 | 2 | 124 | | |
| Income | | | | | | | | | | | 226 | | |
| under 400 € | 12 | 10 | 7 | 13 | 4 | 7 | 2 | 1 | 1 | 1 | 58 | | |
| 401–800 € | 8 | 12 | 13 | 20 | 9 | 5 | 2 | 1 | 1 | 0 | 71 | 0.466 | 62.783 |
| 801–1200 € | 4 | 6 | 15 | 7 | 3 | 0 | 1 | 2 | 1 | 2 | 41 | (0.074) [a] | (0.074) [a] |
| 1201–1600 € | 1 | 6 | 4 | 7 | 4 | 0 | 3 | 1 | 0 | 0 | 26 | | |
| more than 1600 € | 1 | 3 | 6 | 11 | 4 | 3 | 0 | 0 | 2 | 0 | 30 | | |

[a] Significance for C and $\chi^2$ calculation is higher than $\chi^2$ theoretical; in these case $p > 0.05$.

### 3.2. Consumer Perception of Healthy Produce According to Demographics

The above analysis shows that regardless of gender, age, education, and income, the respondents emphasize the nutritional quality and the absence of synthetic chemicals in organic production (Table 3).

**Table 3.** Consumer perception to the health of organic products.

| Variable and Level | Less Healthy | Just Healthy | Healthier | Total | C (Sign) | $\chi^2$ (Sign) |
|---|---|---|---|---|---|---|
| Gender | | | | 226 | | |
| male | 3 | 15 | 100 | 118 | 0.115 (0.217) [a] | 3.052 (0.217) [a] |
| female | 8 | 15 | 85 | 108 | | |
| Age | | | | 226 | | |
| 20–29 years | 3 | 6 | 67 | 76 | | |
| 30–39 years | 4 | 15 | 60 | 79 | 0.140 (0.337) [a] | 4.548 (0.337) [a] |
| 40–49 years | 4 | 9 | 58 | 71 | | |
| Educational level | | | | 226 | | |
| not education | 0 | 0 | 3 | 3 | | |
| primary/second school | 0 | 1 | 2 | 3 | | |
| vocational school | 4 | 6 | 16 | 26 | 0.231 (0.122) [a] | 12.723 (0.122) [a] |
| high school | 3 | 10 | 57 | 70 | | |
| university study | 4 | 13 | 107 | 124 | | |
| Income | | | | 226 | | |
| under 400 € | 1 | 7 | 50 | 58 | | |
| 401–800 € | 4 | 11 | 56 | 71 | | |
| 801–1200 € | 4 | 4 | 33 | 41 | 0.154 (0.702) [a] | 5.506 (0.702) [a] |
| 1201–1600 € | 1 | 5 | 20 | 26 | | |
| more than 1600 € | 1 | 3 | 26 | 30 | | |

[a] Significance for C and $\chi^2$ calculation is higher than $\chi^2$ theoretical; in these case $p > 0.05$.

The level of education of the respondents does not significantly influence the responses, regardless of whether the item factor calculated $\chi^2$ is higher, compared to the theoretical value, which means that the responses are not dependent on the level of education.

When it comes to income, there is a segmentation of consumer perceptions towards organic products, confirmed by the expected value of the $\chi^2$, where $p > 0.05$. Generally, the people with a lower income (86.21%) and those with the highest income (86.67%) believe that organic vegetable products are healthier and richer in nutrients compared to people with an income between 800–1600 €.

Regardless of income, a relatively large number of the respondents (13.27%) consider that organic products are just as healthy as conventional ones, and 4.86% of the respondents believe that organic products are less healthy than conventional ones. In this category, the most negative responses were given by the people with an average income between 400 € and 1200 €.

Depending on the level of education, the people surveyed said that the answers vary depending on the category of the respondents. The people without an education or more than eight years of school attach a great importance to the health of organic products at over 66.67%. The people with technical skills deemed that organic products should be valued primarily for the taste and then for the decreased content of synthetic chemicals. The fact that organic products are healthier and have a good taste is the main criterion for a positive perception for the people with a high school education. The people with higher level of education, besides the nutritional quality issues, consider that the method to obtain these products is consistent with the protection of the environment. This is very important because, in addition to a healthy consumption of products, we would do well for ourselves and future generations, without disturbing, in a negative sense, the balance of the agro-system. The answers achieved nationwide have the same meaning, with responses obtained at an EU level, which shows a concern for consumers to protect the ecosystem [32–34].

Not to be overlooked is the aspect that only 10.49% of cases in the people with a higher education do not know the differences between organic and conventional products, which means that they

had never used or did not have enough information regarding this product. The category of people undecided, or the layman, may represent an available segment for the marketers to exploit in the future.

Regarding consumer perception towards the statement "the organic products are healthier", we did not observe a dependence of the responses on gender, age, and education, with the exception of income level where, $p > 0.05$.

Pieniak [35] and Stolz et al. [36] found that household income did not influence a willingness to pay for organic food, compared to Hjelman [25] and Jörgensen [37], where the expectation of households with middle and higher income levels was a tendency to buy organic produce.

### 3.3. Consumer Perception of Taste Products According to Demographics

The taste of organic products is another criterion among the consumers, regardless of social class or income appreciation for organic products (Table 4). The results of the analysis were assessed by the chi-square coefficient ($\chi^2$) and the contingency coefficient for $p > 0.05$. In order to assess the taste of organic products, three items for the responses were determined: Less tasty, tasty, and tastier than conventional products.

**Table 4.** Perception to products' taste of organic vs. conventional.

| Variable and Level | Less Tasty | Just of Tasty | More Tasty | Total | C (Sign) | $\chi^2$ (Sign) |
|---|---|---|---|---|---|---|
| Gender | | | | 226 | | |
| male | 15 | 25 | 78 | 118 | 0.121 (0.189) [a] | 0.335 (0.189) [a] |
| female | 16 | 33 | 59 | 108 | | |
| Age | | | | 226 | | |
| 20–29 years | 10 | 24 | 42 | 76 | | |
| 30–39 years | 15 | 15 | 49 | 79 | 0.162 (0.193) [a] | 6.083 (0.193) [a] |
| 40–49 years | 6 | 19 | 46 | 71 | | |
| Educational level | | | | 226 | | |
| no education | 0 | 1 | 2 | 3 | | |
| primary/second school | 0 | 2 | 1 | 3 | | |
| vocational school | 4 | 7 | 15 | 26 | 0.174 (0.527) [a] | 7.091 (0.527) [a] |
| high school | 13 | 20 | 37 | 70 | | |
| university study | 14 | 28 | 82 | 124 | | |
| Income | | | | 226 | | |
| under 400 € | 11 | 19 | 28 | 58 | | |
| 401–800 € | 12 | 13 | 46 | 71 | | |
| 801–1200 € | 3 | 12 | 26 | 41 | 0.197 (0.329) [a] | 9.165 (0.329) [a] |
| 1201–1600 € | 3 | 7 | 16 | 26 | | |
| more than 1600 € | 2 | 7 | 21 | 30 | | |

[a] Significance for C and $\chi^2$ calculation is higher than $\chi^2$ theoretical; in these case $p > 0.05$.

From the total number of the respondents, 137 responses were positive in that they appreciated the taste, indifferent by gender.

From the pronounced answers given from the perception of the consumer with the object of organic products we observed that, according to the respondents' age, over 55% of the respondents believe that products are healthy. A high percentage of the respondents (25.66%) believe that products are just as tasty as conventional ones, including most of the answers in the interview that were given by the age group 20–29 years, which, in marketing terms, is as an age segment to be exploited and educated. The respondents aged 30–39 years considered, in the proportion of 18.98%, that products are less tasty than conventional ones. The level of education plays a major role in the segmentation of the respondents' perception of organic products, confirmed by the $\chi^2$ and C coefficients in which $p > 0.05$. In 60.62% of the cases, the respondents believe that organic products are tasty. Among them, most answers were given by people with higher education (66.13%). The consumers with a high school level of education provided a segmentation of the responses from the three categories of items. Thus, 47.15% of them consider that organic products are less or just as tasty as conventional ones.

Depending on income, over 60% of the respondents believe that organic vegetables are tasty. A higher segmentation of perception is observed from the people with incomes under 400 €, where 48.28% believe that organic products are less tasty or just as tasty as conventional products.

Regardless of income, age, and gender, the respondents considered that organic products are just as tasty, with a percentage of 60.62%. However, the level of training influences the perception of the consumer ($p > 0.05$) meaning that the people with no training and with no more than eight years of school believe that organic products are as tasty, the people with high school education think that the organic products are less tasty or as tasty as the conventional (47.15%), and those with higher education, with a percentage of 66.13%, consider organic products to be just as tasty; there are also people (22.85%) who believe that conventional vegetables are just as tasty as organic ones.

### 3.4. Consumer Perception of the Appearance of Produce According to Demographics

Consumer perception towards organic rather than conventional products was assessed by three items. In this case, statistically, the responses obtained were assessed by the chi-square coefficient ($\chi^2$) values and the contingency coefficient (C) to determine the dependence of the demographic factor and the Fisher test (F) for the mean sample.

The quality of the products, due to a low content of chemical synthetic substances, can be assessed with other sensory properties, such as appearance, flavor, aroma, and texture.

In the minds of the consumers, in terms of gender, the significance for $\chi^2$ can conclude that the appreciation of the layout does not depend on the respondents' gender. The respondents' perception is an element of segmentation and the dependence of the responses to the appearance of organic products in the situation when $p > 0.05$. The respondents, with a percentage of 36.73%, believe that organic products are less appealing, 42.48% believe that they are as appealing as conventional ones, and relatively low proportions consider the conventional products to be inferior to the organic ones in terms of appearance (Table 5). In the 20–29 age group, 85.53% of the respondents believe that organic products have less-appealing appearances then conventional products.

**Table 5.** Consumers' perception than organic vegetable appearance.

| Variable and Level | Less Appearance | Just of Appearance | Much Appearance | Total | C (Sign) | $\chi^2$ (Sign) |
|---|---|---|---|---|---|---|
| Gender | | | | 226 | | |
| male | 43 | 46 | 29 | 118 | 0.103 (0.299) [a] | 2.412 (0.299) [a] |
| female | 40 | 50 | 18 | 108 | | |
| Age | | | | 226 | | |
| 20–29 years | 32 | 33 | 11 | 76 | | |
| 30–39 years | 27 | 32 | 20 | 79 | 0.121 (0.500) [a] | 3.356 (0.500) [a] |
| 40–49 years | 24 | 31 | 16 | 71 | | |
| Educational level | | | | 226 | | |
| not education | 0 | 3 | 0 | 3 | | |
| primary/secondary school | 2 | 1 | 0 | 3 | | |
| vocational school | 7 | 13 | 6 | 26 | 0.181 (0.465) [a] | 7.686 (0.465) [a] |
| high school | 27 | 26 | 17 | 70 | | |
| university study | 47 | 53 | 24 | 124 | | |
| Income | | | | 226 | | |
| under 400 € | 25 | 24 | 9 | 58 | | |
| 401–800 € | 28 | 23 | 20 | 71 | | |
| 801–1200 € | 12 | 24 | 5 | 41 | 0.225 (0.150) [a] | 12.035 (0.150) [a] |
| 1201–1600 € | 8 | 10 | 8 | 26 | | |
| more than 1600 € | 10 | 15 | 5 | 30 | | |

[a] Significance for C and $\chi^2$ calculation is higher than $\chi^2$ theoretical; in these case $p > 0.05$.

In the 30–39 years age group, the segmentation of the answers to the three categories show that 34.18% of the respondents say that organic vegetables are less appealing; 40.5% consider that there are no differences in terms of appearance according to the appearance between organic products

and conventional ones, and 25.32% consider that organic products are much more appealing. In the age group of 40–49 years, the respondents believe that organic products are just as appealing as conventional ones (43.67%) which is close to the average of the entire sample of 42.48%. It can be concluded that the survey sample has a high percentage of over 42% to be exploited positively to eati organic products, corroborating the look with the taste as well as the high quality of these products. The same results were shown through scientific literature [36,38].

The appearance of organic vegetable products is a perception widely available, depending on the level of education. Regardless of the level of training, most of the interviewed persons considered organic products to be less appealing or just as appealing as conventional products. The most favorable responses, in this way, were given by the people without a formal education or with a primary school level (100%) and the persons with higher education (80.65%). From a total, 47 of the respondents believe that organic products have a more pleasing appearance according to the aspect, the same found by Bryla [39].

In the sample, most of the respondents believe that organic products are less appealing (36.72%) and just as appealing (42.47%). The homogeneous responses were observed in individuals with an income in the range of 600–1200 €.

Appearance, as a factor in the perception of consumer sensory, indicates that in the sample, regardless of gender, age, level of education, and income, 36.73% of the people believe that organic products are less appealing. In 42.48% of the cases, the respondents considered that organic products could not be distinguished from conventional ones in terms of appearance, and only 20.80% of the respondents thought that the organic products were appealingly pleasant. The considerations for the organic vegetable product regarding its health benefits and lack of chemicals in accordance with friendly technology toward the environment tend to emphasize that the respondents believe that these products are not as appealing as conventional produce. Also, the consumption of organic products can be in terms of their quality. The health benefits of organic food are required as a primary way for the organic consumer [27,40,41].

Depending on age, it appears that the issue is a criterion of dependency between the two factors, where $p > 0.05$. Thus, 42.11% of the people in the 20–29 years old group believe that organic products are even less welcome than the 40–49 years old group; their responses were 33.81%. This is explained by the fact that young people generally buy ready-made food, and many of them are students living with their parents, and the latter certainly buy fresh products and have more information from the media. At the opposite end, the segment of middle age persons, from a percentage of 25.32%, think that organic products are more appealing; they could be people who purchase their foods, usually, from supermarkets and less from the domestic market, where the requirement is higher.

Research done over the course of time has identified a number of socio-economic and demographic variables that has significantly influenced demand or a willingness to pay for organic goods. Misra et al. [32] found no significant correlation between age and buying propensity.

Buzby and Skees [42] and Zind [43] found that younger consumers (under 45 years old) have a higher propensity to purchase organic products than older consumers. However, other studies have found [33,40] that older consumers also have a high propensity to buy organic products.

## 4. Conclusions

The urban consumer's perception of organic products is not significantly influenced by age, gender, income, or education level. Educated and non-educated urban consumers have specific knowledge about organic products. Approximately half of respondents associate the high nutritional quality of bio-products with low pesticide residues from food.

Most respondents consider organic products to be healthier and richer in nutrients. Nutritional quality and sensory attributes (appearance, taste, flavor) are associated with organic technology. The chemical composition and the functional properties are consistent with the relationship to the

environment. Certain criteria, like the price, a lack of GMO, or the label are not currently perceptual factors that consumers from Romania must take into account.

The reduced consumption of chemicals in organic farming is the main criteria for which the consumers choose products. When it comes to the respondents' perception of sensory quality of the organic products, it can be said that a majority of the respondents consider organic products less appealing but instead tastier. The results of the study show a positive consumer perception for the taste of the organic products, indifferent for the level of education. Educated consumers believe that the sensory quality of organic products is its taste, in over two-thirds of the cases, compared to those with a high school or technical level of education, where the responses were about half.

Promoting the idea of healthy-looking food when it comes to organic farming will positively shape the concept of the perception. The products differ in the appearance category, by the fact that there is a lack of growth hormones, synthetic fertilizers, and pesticides; the products are usually smaller, with imperfections, and come in different shapes and colors, etc.

The results of the research are important because they provide factual information to consumers from major urban centers in Romania, but the information is practical for both organic producers and organic traders, because it creates channels and niches for production and selling.

**Author Contributions:** Conceptualization, V.S. and N.M.; methodology, N.M. and V.S.; investigation, A.I. and V.S.; data analysis, A.I.; writing—original draft preparation, V.S. and A.I.; writing—review and editing, V.S. and N.M.

**Funding:** This research received funding by EU PROJECT MIS-ETC 927.

**Acknowledgments:** We thank Dean Hufstetler for revised English.

**Conflicts of Interest:** The authors declare no conflict of interest.

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
