# Peer review of "Perception Towards Organic vs. Conventional Products in Romania"

_sustainability, doi:10.3390/su11082394_

Round 1

Reviewer 1 Report

The Authors made an effort to better clarify the objective and improve the introduction. However, I still have the same main concern as before from the first draft. How perceptions were measured?

In the method section, you mentioned

The first group of questions was aimed at identifying the perceived differences between organic and conventional produce such as:

-Compared to conventional products, organic products are: less tasty, just as tasty, more tasty, less flavorful, just as flavorful, more flavorful, fast breaks, spoil as quickly, less healthy, just as healthy, healthier, less appealing, appealing, more appealing;

But how you measured the difference?. It is not clear so the Analysis done are not clear (ANOVA or Chi-Square?). was the difference measured only by “yes” or “No” option? Did you use an agreement scale? 7 or 9 points Likert Type Scale?

The explanation of the questions about the difference in perception from line 143 until line 157 needs to be improved.  Again Clarify how the different concepts were measured?

How the authors expect from readers to understand the Tables in the results section without mentioning before  how you measured each variable? How is possible to apply ANOVA and Chi square for the same variables test. Put the mean when using ANOVA.....

For instance, How is it possible in Table 2, to see the  ANOVA and Chi square for the same Variables tested¡¡¡¡?. Too confusing… if you are comparing the means for more than two groups (like education, income and age categories) the ANOVA is not enough. You should carry out also post hoc analysis such as Tukey.

The same also occurs with all remaining Table. I was not able to understand the analysis done because how perceptions were measured is not clear.

Author Response

In the method section, you mentioned

The first group of questions was aimed at identifying the perceived differences between organic and conventional produce such as:

-Compared to conventional products, organic products are: less tasty, just as tasty, more tasty, less flavorful, just as flavorful, more flavorful, fast breaks, spoil as quickly, less healthy, just as healthy, healthier, less appealing, appealing, more appealing;

But how you measured the difference?. It is not clear so the Analysis done are not clear (ANOVA or Chi-Square?). was the difference measured only by “yes” or “No” option? Did you use an agreement scale? 7 or 9 points Likert Type Scale?

Population sampling is largely determined by a certain comfort, people were randomly selected from the main urban centers to exit the supermarkets, knowing that urban people are the main consumers of organic products, so they have perceptions of food.

The presentation of the methodology has been restored. The questionnaire consisted of 14 problems, in three sections: (1) filtering questions, (2) differences in perception between organic and conventional foods (common), and (3) socio-professional characteristics. The first part of the questionnaire focused on sample filters like gender and age of the respondent.

The second group of questions was aimed at identifying the perceived differences between organic and conventional produce such as:

-compared conventional products with organic products are: less tasty, just as tasty, tastier, less flavorful, just as flavorful, more flavorful, fast breaks, spoil as quickly, less healthy, just as healthy, healthier, less appealing, appealing, more appealing. At questions about consumer perception regarding health, taste and appearance, was used a Likert scale with three response options (Less .... = 1; Just of .... = 2; More .... = 3), created a homogeneous symmetry between the three types of response.

- differences perceived by consumers between organic and conventional products to: content of chemicals, healthy, environmental protection, price, preservation, label, appearance, absence of GMOs etc. At this question, consumers had to choose the most important criterion for differentiating between organic and conventional products.

To reduce error as far possible, the sampling frame covering the gender and age, according to the demographic profile.

Consumer age is a determined factor in perceiving organic consumption. Choosing the 3 age categories is an important demographic factor in creating perceptions, knowing that people in the 20-49 age category are responsible for choosing food, are engaged people with a good economic situation, are people who buy and cook away for children and the elderly.

The explanation of the questions about the difference in perception from line 143 until line 157 needs to be improved. Again Clarify how the different concepts were measured?

The questions provided by line 143 to 157 have been modified so that only those variables remained strictly on the field of study.

How the authors expect from readers to understand the Tables in the results section without mentioning before  how you measured each variable? How is possible to apply ANOVA and Chi square for the same variables test. Put the mean when using ANOVA.....

For instance, How is it possible in Table 2, to see the ANOVA and Chi square for the same Variables tested¡¡¡¡?. Too confusing… if you are comparing the means for more than two groups (like education, income and age categories) the ANOVA is not enough. You should carry out also post hoc analysis such as Tukey.

The same also occurs with all remaining Table. I was not able to understand the analysis done because how perceptions were measured is not clear.

Statistical analyses was restore, can see on methodology and tables of data analyses (tab. 2,3,4,5)

Data was quantitatively collected and analyzed using the SPSS software version 20 from IBM. A variety of descriptive and inferential statistics were utilized to understand significant differences in consumer perception according to the demographics for specific attributes.

The ANOVA test was utilized to summarize the demographic profile. Since the data are also unanswered variables or the number of responses is less than 2, the statistical analysis were made by Contingency Test (C) and Chi-square Pearson test (χ2), thus demonstrating the dependence of the variables, or the probability that any observed difference between variables will occur by chance.

Add.

References and the model of citation was remade.

All changes in text are made by red color.

Reviewer 2 Report

The subject area discussed in the paper is important. It is also consistent with the profile of the Journal. Deliberations conducted in the paper need to be expanded.

To improve the quality of the paper I would suggest to:

- develop description concerning methodology of conducted research and criteria of respondents’ recruitment,

- specify the interpretation of obtained results of the research and managerial implications, as the considerations in these parts of the paper are not thorough enough, 

- provide in-depth and expanded conclusions,

- formulate trends for future research.

Author Response

To improve the quality of the paper I would suggest to:

- develop description concerning methodology of conducted research and criteria of respondents’ recruitment,

Population sampling is largely determined by a certain comfort, people were randomly selected from the main urban centers to exit the supermarkets, knowing that urban people are the main consumers of organic products, so they have perceptions of food.

To reduce error as far possible, the sampling frame covering the gender and age, according to the demographic profile.

Consumer age is a determined factor in perceiving organic consumption. Choosing the 3 age categories is an important demographic factor in creating perceptions, knowing that people in the 20-49 age category are responsible for choosing food, are engaged people with a good economic situation, are people who buy and cook away for children and the elderly.

The presentation of the methodology has been restored. The questionnaire consisted of 14 problems, in three sections: (1) filtering questions, (2) differences in perception between organic and conventional foods (common), and (3) socio-professional characteristics. The first part of the questionnaire focused on sample filters like gender and age of the respondent.

The second group of questions was aimed at identifying the perceived differences between organic and conventional produce such as:

-compared conventional products with organic products are: less tasty, just as tasty, tastier, less flavorful, just as flavorful, more flavorful, fast breaks, spoil as quickly, less healthy, just as healthy, healthier, less appealing, appealing, more appealing. At questions about consumer perception regarding health, taste and appearance, was used a Likert scale with three response options (Less .... = 1; Just of .... = 2; More .... = 3), created a homogeneous symmetry between the three types of response.

- differences perceived by consumers between organic and conventional products to: content of chemicals, healthy, environmental protection, price, preservation, label, appearance, absence of GMOs etc. At this question, consumers had to choose the most important criterion for differentiating between organic and conventional products.

- specify the interpretation of obtained results of the research and managerial implications, as the considerations in these parts of the paper are not thorough enough, 

Data was quantitatively collected and analyzed using the SPSS software version 20 from IBM. A variety of descriptive and inferential statistics were utilized to understand significant differences in consumer perception according to the demographics for specific attributes.

The ANOVA test was utilized to summarize the demographic profile. Since the data are also unanswered variables or the number of responses is less than 2, the statistical analysis were made by Contingency Test (C) and Chi-square Pearson test (χ2), thus demonstrating the dependence of the variables, or the probability that any observed difference between variables will occur by chance.

- provide in-depth and expanded conclusions,

The urban consumer’s perception of organic products is not significantly influenced by age, gender, income or education level. Educated and non-educated urban consumers, have specific knowledge about organic products. Approximately half of respondents associate the high nutritional quality of bio-products with low pesticide residues from food.

The results of the study show a positive consumer perception for the taste of the organic products, indifferent of the level of education.

- formulate trends for future research.

The results of the research are important because they provide factual information to consumers from major urban centers in Romania, but the information is practical for both organic producers and organic traders, because it creates channels and niches for production and selling.

Starting from this research, new opportunities can be developed for consumers' preferences towards organic products, or in universities, studies can be made on the behavior or attitudes of younger generations towards organic farming technology in line with environmental principles.

The results of the study are addressed to all decision-makers (consultancy centers, schools, universities) who, through their activities, can determine, educate and inform consumers to have a solid knowledge of organic products perception.

The study is useful for teachers who teach marketing and food discipline to students in the field. Also, such research is useful for traders to develop some niches for the capitalization of organic products.

Add.

References and the model of citation was remade.

All changes in text are made by red color.

Round 2

Reviewer 1 Report

It is ok for me. Now the authors clarified the statistics done. In all cases, all results in all tables were not significant which puts in evidence the methodological approach followed. This could be related to the very small interval used in their variables (only 3 category: less… just off…. more).

This manuscript is a resubmission of an earlier submission. The following is a list of the peer review reports and author responses from that submission.

Round 1

Reviewer 1 Report

Introduction

Line 27: update data.

Line 39-41 who stated these affirmations. It should be referenced to literature or Relay on well known consumers’ theory.

Line 52: Before purchase happens, people must be aware of the product. This is not necessary always, in particular when new products are offered at market place and consumer have not experience with it or knowledge. Before purchase many attributes of a food product cannot be revealed, so consumers develop expectations about quality when selecting it. In this case Expectations of new products are framed by the information on the label and previous experience with a similar product (Kallas, et al., 2019).

Kallas, Z., Vitale, M., & Gil, J. M. (2019). Health Innovation in Patty Products. The Role of Food Neophobia in Consumers’ Non-Hypothetical Willingness to Pay, Purchase Intention and Hedonic Evaluation. Nutrients, 11(2), 444.

In all cases, the introduction is confusing with several ideas that are not well connected with a common discourse. Therefore, the objective seems too simple and not based on what the authors commented in the introduction. There is unclear added-value of this research. What is the contribution of the paper compared to literature of consumer preference towards organic food? And the main take-home message? Thus if the objective is to analyze perception wht the title focused on attitude?

Line 89 structured according to demographics should be stratified by…

Sample should be better explained. Which sampling method and error term?. Did you present them a Consent form to accept it? Currently the EU legislation in social science studies make a specific care on protecting personal information according to the European General Data Protection Regulation No 2016/679. Participants should receive an explanation of the objective of the study, making emphasis that the information requested will be exclusively used for research and confidentiality is absolutely guaranteed. Furthermore, should be informed that their participation is voluntary and they were randomly selected to participate. Please clarify this point.

data collection was face to face or on line?

Questionnaire. Provide information if the document was also approved by ethical committee in social science. Provide also the questionnaire used.

Table 1 not necessary

Have you verified if ANOVA as parametric test is applicable? Did you test for normality of the variables?

If the objective is also to segment the sample, why you did not carry out a cluster analysis?

The English need a deep review. Several expressions are not clear

Reviewer 2 Report

In the assessment of the paper submitted for the review, I specifically focused on the discussed issues, applied research procedure, substantive content of the paper and its structure.

The considerations conducted in the paper are focused on such categories as: attitude, consumer behavior, organic products, conventional products.

The subject area discussed in the paper should be considered important and consistent with the profile of the Journal. The paper is based on literature studies and results of an empirical research, which was conducted on a sample of 226 respondents in 2013 year in Romania.

Deliberations conducted in the paper need to be expanded. Therefore, it is specifically recommended to:

- develop description concerning methodology,

- take into consideration other latest publications in the sphere of discussed subject matter,

- expand the scope of interpretation of the results of empirical research,

- indicate the research limitations and formulate trends for future research (as a separate part of article),

- formulate managerial implications.